# The Lower Limb Muscle Co-Activation Map during Human Locomotion: From Slow Walking to Running

**DOI:** 10.3390/bioengineering11030288

**Published:** 2024-03-19

**Authors:** Lorenzo Fiori, Stefano Filippo Castiglia, Giorgia Chini, Francesco Draicchio, Floriana Sacco, Mariano Serrao, Antonella Tatarelli, Tiwana Varrecchia, Alberto Ranavolo

**Affiliations:** 1Department of Occupational and Environmental Medicine, Epidemiology and Hygiene, INAIL, Via Fontana Candida 1, Monte Porzio Catone, 00078 Rome, Italy; g.chini@inail.it (G.C.); fdraicch@gmail.com (F.D.); fl.sacco@inail.it (F.S.); antonellatatarelli@gmail.com (A.T.); t.varrecchia@inail.it (T.V.); a.ranavolo@inail.it (A.R.); 2Behavioral Neuroscience PhD Program, Department of Physiology and Pharmacology, Sapienza University, Viale dell’Università 30, 00185 Rome, Italy; 3Department of Medical and Surgical Sciences and Biotechnologies, Sapienza University of Rome, Polo Pontino, Via Franco Faggiana 1668, 04100 Latina, Italy; stefanofilippo.castiglia@uniroma1.it (S.F.C.); mariano.serrao@uniroma1.it (M.S.); 4Department of Brain and Behavioral Sciences, University of Pavia, Via Bassi 21, 27100 Pavia, Italy

**Keywords:** muscle co-activations, walking, running, spinal map, sEMG

## Abstract

The central nervous system (CNS) controls movements and regulates joint stiffness with muscle co-activation, but until now, few studies have examined muscle pairs during running. This study aims to investigate differences in lower limb muscle coactivation during gait at different speeds, from walking to running. Nineteen healthy runners walked and ran at speeds ranging from 0.8 km/h to 9.3 km/h. Twelve lower limb muscles’ co-activation was calculated using the time-varying multi-muscle co-activation function (TMCf) with global, flexor–extension, and rostro–caudal approaches. Spatiotemporal and kinematic parameters were also measured. We found that TMCf, spatiotemporal, and kinematic parameters were significantly affected by gait speed for all approaches. Significant differences were observed in the main parameters of each co-activation approach and in the spatiotemporal and kinematic parameters at the transition between walking and running. In particular, significant differences were observed in the global co-activation (CI*_glob_*, main effect F_(1,17)_ = 641.04, *p* < 0.001; at the transition *p* < 0.001), the stride length (main effect F_(1,17)_ = 253.03, *p* < 0.001; at the transition *p* < 0.001), the stride frequency (main effect F_(1,17)_ = 714.22, *p* < 0.001; at the transition *p* < 0.001) and the Center of Mass displacement in the vertical (CoM*_y_*, main effect F_(1,17)_ = 426.2, *p* < 0.001; at the transition *p* < 0.001) and medial–lateral (CoM*_z_*, main effect F_(1,17)_ = 120.29 *p* < 0.001; at the transition *p* < 0.001) directions. Regarding the correlation analysis, the CoM*_y_* was positively correlated with a higher CI*_glob_* (*r* = 0.88, *p* < 0.001) and negatively correlated with Full Width at Half Maximum (FWHM*_glob_*, *r* = −0.83, *p* < 0.001), whereas the CoM*_z_* was positively correlated with the global Center of Activity (CoA*_glob_*, *r* = 0.97, *p* < 0.001). Positive and negative strong correlations were found between global co-activation parameters and center of mass displacements, as well as some spatiotemporal parameters, regardless of gait speed. Our findings suggest that walking and running have different co-activation patterns and kinematic characteristics, with the whole-limb stiffness exerted more synchronously and stably during running. The co-activation indexes and kinematic parameters could be the result of global co-activation, which is a sensory-control integration process used by the CNS to deal with more demanding and potentially unstable tasks like running.

## 1. Introduction

Muscle co-activation is one of the strategies used by the central nervous system (CNS) to simplify movements and ensure adequate joint stiffness by regulating the time and amplitude of simultaneous activity of a pair or group of muscles [1,2,3,4]. Muscle co-activation is thought to maintain effector-level control (low dimensional), removing the need for individual muscle coordination control (high dimensional) [2].

Although muscle co-activation has been extensively studied during gait in both healthy patients and patients with a variety of motor disorders (for a review, see [2]), only a few studies have looked at muscle co-activation during running [5,6,7,8,9]. Walking to running represents a critical transition time that corresponds to a change in CNS activation [10], leg geometry [11,12], joint compliance [13], and a robust mechanical energy transformation [14,15,16]. Indeed, walking and running patterns, despite the similarity, differ in spatiotemporal and kinematic characteristics, such as the lower limb joint angles [17,18].

Most humans will voluntarily switch from walking to running at 6.8 and 7.5 km/h [19,20,21,22]. At these higher speeds, running becomes less expensive than walking by utilizing a mass-spring mechanism that exchanges kinetic and potential energy in very different ways [15,23]. Most studies on muscle activation during running have concentrated on the activity of single muscles [24,25,26]. Running has been linked to the increased activation of all lower limb muscles in general [5,27,28,29], despite the fact that some muscles (e.g., gluteus maximus) appear to play a larger role during running than walking [30]. Only a few studies have looked at the activation of muscle pairs while running [9,31]. When compared to a single-joint muscle solution, these studies discovered that a longer duration of muscle co-activation between the rectus femoris and gastrocnemius during stance provided a better metabolic solution to multiple joint stability, implying that biarticular muscles redistribute mechanical power from proximal joints to distal joints, and ultimately to the ground. Moore et colleagues [9] discovered that co-activation in the distal and leg flexor muscles decreases with speed while running, implying that when both legs are off the ground, muscle co-activation must be reduced to allow for more leg propulsion both upwards and forwards.

A critical missing piece is whether the descending motor commands for running are set up to coactivate the lower limb muscles as a whole entity within a gross motor strategy. Taking previous evidence into account [9,31], it is possible to hypothesize that descending commands for running cause an increase in whole limb stiffness to stabilize the limb during ground contact on the one hand and a decrease in limb stiffness to facilitate forward progression of the leg during air stepping on the other. This basic descending motor control would imply that flexor muscles are activated during air stepping when whole-limb stiffness is reduced and that muscle co-activation occurs via top-down (rostro–caudal) recruitment (from L3 to S2). This strategy would greatly simplify motor control at the low-dimensional effector level during shock absorption, allowing for motor control decentralization while relying on peripheral feedback via muscle reflexes to generate the required muscle co-activations and regulate each muscle’s gain.

Muscle co-activation patterns are determined by the supraspinal processes involved in movement control [2]. Two main circuits, corticocerebellar–thalamo–cortical and corticobasal–thalamo–cortical, appear to be important in defining co-activation patterns [32]. In particular, studies of subjects with cerebellar degeneration [33], spasticity [34], or extrapyramidal rigidity [35] have found that the cerebellum plays a role in modulating muscle co-activation, implying that processing modules distributed within the brain define the patterns of co-activation observed in the extremities [36].

We recently proposed a novel approach to studying time-varying multi-muscle co-activation function (TMCf) [37,38], which is a good indicator of the CNS’s overall strategy for modulating the simultaneous activation of many lower limb muscles during locomotion. This approach provides a new perspective on the spatiotemporal motor control of the lower limbs, emphasizing how all muscles of the lower limb are synchronously coactivated in an attempt to increase whole-limb stiffness, regardless of single-joint antagonist muscles or modular activation of a group of muscles [39].

The objectives of this study were to observe the differences in lower limb muscle co-activation strategies during locomotion across speeds from walking to running. We hypothesized that the behavior of lower limb muscles, in terms of co-activation parameters, may differ across the speeds, particularly between walking and running, and that the co-activation parameters may vary based on the muscular functions (flexion or extensor functions) and across the rostro–caudal recruitment map.

The findings of this study may aid clinicians to better understand the mechanisms of muscular behavior during the transition from walking to running, potentially providing information on control abnormalities that lead to muscle injuries, as well as sport professionals in developing tailored training programs.

## 2. Materials and Methods

### 2.1. Subjects

Nineteen healthy runners were recruited (9 men and 10 women; mean age of 40.95 ± 7.96 years; mean weight of 66.47 ± 14.60 kg). Each runner had declared running for at least 5 years, running at least 3 times per week for at least 5 km per training session [40].

None of the runners had any known diseases that might affect their regular gait or running pattern. This study was authorized by the local ethics committee (N. 0078009/2021) after all participants gave written informed consent and the study’s design adhered to the Declaration of Helsinki. The number of the included subjects was in the range of subjects usually enrolled in similar studies [41,42,43]. Moreover, a priori power analysis using the G∗Power computer program [44] indicated that a total sample of 14 participants would be needed to detect a medium effect size (Cohen’s F = 0.25) with 80% power using the repeated measurement ANOVA with gait speed as a within-subjects factor with a = 0.05.

### 2.2. Experimental Procedure

Each runner was asked to walk at thirteen various speeds, ranging from 0.8 km/h to 6.8 km/h, and run at five different speeds, ranging from 7.3 km/h to 9.3 km/h, with increased steps of 0.5 km/h, on a treadmill. We chose 7.3 km/h as the transition speed because it is consistent with earlier research [19,21,22] in which this transition was determined considering the metabolic energy cost of locomotion, as well as the human capacity for purposeful gait modulation and the importance of physiologic and metabolic demands.

Before the recording session, individual speed performance schedules were arranged for each participant, featuring randomized presentations. Each runner underwent a 10 s adaptation period to familiarize with the given speed, followed by a 30 s recording period. Following each trial, the speed was reset to zero, and the runner was granted a minimum of 1 min of stationary rest to prevent fatigue before commencing the subsequent trial. Each trial was performed for 30 s before moving on to the next speed that was chosen at random.

### 2.3. Data Acquisition

A bipolar 16-channel wireless acquisition device (Mini Wave System; Cometa, Bareggio, Milan, Italy) was used to record all the surface myoelectric activity at a sampling rate of 2 kHz. Twelve Ag/AgCl pre-gelled electrodes (Kendall ARBO, inter-electrode distance: 2 cm) were placed over each participant’s right side lower limb muscles over the following muscles: gluteus medius (GM), rectus femoris (RF), vastus lateralis (VL), vastus medialis (VM), tensor fascia latae (TFL), semitendinosus (ST), biceps femoris (BF), tibialis anterior (TA), gastrocnemius medialis (GasM), gastrocnemius lateralis (GasL), soleus (SOL), and peroneus longus (P) [45,46], following the European Recommendations for Surface Electromyography [47], the Atlas of Muscle Innervation Zones [48], and best practices [45,46,49]. Before applying the electrodes, the skin was prepared by shaving, if needed, and cleaned with alcohol and dried.

A stereo-photogrammetric motion analysis system with optoelectronic technology (SMART-DX 6000 system: BTS, Bari, Italy) and with a sampling rate of 340 Hz was employed to collect the kinematics data. Eight infrared cameras were used to record five passive spherical markers, covered with aluminum powder, positioned above the sacrum and bilaterally on the anterior superior iliac spines, heel, metatarsal head, and lateral malleoli [50].

A video camera (BTS Vixta; BTS, Milan, Italy) with a frame rate of 25–30 frames per second and a video resolution of 640 × 480 pixels was used to visually monitor the acquired tasks. All the data collected were synchronized.

### 2.4. Data Analysis

Three-dimensional reconstruction software (SMART Tracker and SMART Analyzer: BTS, Italy) and MATLAB (R2019b 9.7; MathWorks, Portola Valley, CA, USA) were used to process the sEMG and kinematics data.

#### 2.4.1. Cycle Definition and Temporal Normalization

The heel strike (HS) and toe-off (TO) events were determined in this study along the anteroposterior trajectories through the maximum point of the heel and the minimum point of the metatarsal, respectively. The gait and run cycle was defined as the time between two consecutive HSs of the same leg. Then, the electromyographic and kinematic data were time-normalized to the duration of each cycle and reduced to 201 samples using an interpolation procedure to allow a comparison between different cycles that had different durations [31,37,38,39]. Ten cycles for each gait speed level for each runner were analyzed.

#### 2.4.2. Global, Flexor, Extensor and Rostro–Caudal Co-Activation of Lower Limb Muscles

The raw sEMG signals were visually reviewed to remove any artifact-containing cycles. They were then band-pass filtered with a zero-lag fifth-order Butterworth (20–450 Hz) [51,52] to keep only the signal of interest and then full-wave rectified and low-pass filtered with a zero-lag fifth-order Butterworth (10 Hz) [53]. The elaborated sEMG signals [54,55] of each muscle were amplitude-normalized (0–100%) for each runner in relation to the mean of their three highest peak values detected across all gait cycles and velocities [33,56].

The TMCf was used to calculate the simultaneous activation of the lower-limb muscles based on the processed sEMG signals [3,33,37,38,57,58,59]. The full-wave-rectified, low-pass-filtered, and 0–100% amplitude normalized sEMG signals were used as inputs to this sigmoid-weighted time-dependent co-activation function for the inclusion of multiple muscles during walking and running. This co-activation function’s values ranged from 0 to 100%, and they were calculated as follows:(1)$$TMCf(d(i),i)=C(d(t))\cdot \frac{{\left(\sum_{m=1}^{M}EMG_{m}(i)/M\right)}^{2}}{max_{m=1\ldots M}\left[EMG_{m}(i)\right]}=\left(1-\frac{1}{1+e^{-a\left(d\left(i\right)-b\right)}}\right)\cdot \frac{{\left(\sum_{m=1}^{M}EMG_{m}(i)/M\right)}^{2}}{max_{m=1\ldots M}\left[EMG_{m}(i)\right]}$$
where C(d(t)) is the sigmoid weight reduction coefficient, *M* is the number of muscles considered, and $EMG_{m}(i)$ is the sEMG sample value of the m-th muscle at instant i. The function C(d(t)), ranging between 0 and 1, takes into account, within the exponential function, the constants a and b equal to 12 and 0.5 respectively [37,57]. *d*(*i*) is the mean of the differences between each pair among the 12 muscles ($EMG_{m}(i)$) values at instant i:(2)$$d(i)=\left(\frac{\sum_{m=1}^{M-1}\sum_{n=m+1}^{M}\left|EMG_{m}(i)-EMG_{n}(i)\right|}{L\left(M!/\left(2!\left(M-2\right)!\right)\right)}\right)=\left(\frac{\sum_{m=1}^{M-1}\sum_{n=m+1}^{M}\left|EMG_{m}(i)-EMG_{n}\left(i\right)\right|}{L\left(M\left(M-1\right)/2\right)}\right)$$
where L is the length of the signal and M!/(2(M−2)!) is the total number of possible differences between each pair of *EMG_m_*(*i*).

TMCf(d(i),i) has the following properties: an inverse relationship with the mean of the differences *d*(*i*), i.e., values close to the mean activation of the m(i) muscle sample values considered when d(i) is close to 0, and values close to 0 when d(i) is close to 1. The smaller the differences in muscle activations are, the closer the d(i) values are to 0; the closer the sigmoid-coefficient values are to 1, the closer the TMCf(d(i),i) value is to its mean value. In contrast, the greater the differences in muscle activations, and the higher the d(i) and the lower the sigmoid coefficient, the lower the TMCf(d(i),i) values will be. Data over individual strides was calculated for each runner and speed and then averaged across cycles.

All the acquired muscles were inserted in the calculation of the TMCf to assess global co-activation (TMCf*_glob_*). The co-activation of extensor (TMCf*_ext_*) and flexor (TMCf*_flex_*) muscles were computed (Table 1) by considering the extensor and flexor muscle subgroups made up according to the “Concentric function”; it indicates that the flexor and extensor subgroups include those muscles which contract concentrically (shortening) during the flexion and extension of the joint [60]. The biarticular muscles were considered flexors or extensors based on their proximal function [61,62]. Furthermore, we calculated the TMCf by separating the muscles on the basis of their spinal segment of innervation (Table 1): first, all the muscles innervated by L3, then those by L4, then those by L5, then the one by S1, and finally those innervated by S2; thus rostro–caudal organization (TMCf*_L_*_3_; TMCf*_L_*_4_; TMCf*_L_*_5_; TMCf*_S_*_1_; TMCf*_S_*_2_) [41,60,63,64] was assessed using subgroups of muscles (see Table 1). Muscles were considered flexors or extensors based on their concentric function in the sagittal plane [60].

#### 2.4.3. Co-Activation Parameters

Within the gait cycle, the following parameters were calculated for each map, runner, and speed:(i)the synthetic co-activation index (CI*_glob_*; CI*_ext_*; CI*_flex_*; CI*_L_*_3_; CI*_L_*_4_; CI*_L_*_5_; CI*_S_*_1_; CI*_S_*_2_) which is calculated as the mean value of the TMCf and represents the average of the co-activation level [% co-activation];(ii)the maximum value of the TMCf (Max*_glob_*; Max*_ext_*; Max*_flex_*; Max*_L_*_3_; Max*_L_*_4_; Max*_L_*_5_; Max*_S_*_1_; Max*_S_*_2_) [% co-activation];(iii)the full width at half maximum (FWHM*_glob_*; FWHM*_ext_*; FWHM*_flex_*; FWHM*_L_*_3_; FWHM*_L_*_4_; FWHM*_L_*_5_; FWHM*_S_*_1_; FWHM*_S_*_2_) of the co-activation, which reflects the sum of the time durations within the gait cycle during which the TMCf curve is higher than its half maximum value [% gait cycle];(iv)the center of activity (CoA*_glob_*; CoA*_ext_*; CoA*_flex_*; CoA*_L_*_3_; CoA*_L_*_4_; CoA*_L_*_5_; CoA*_S_*_1_; CoA*_S_*_2_) which is calculated with circular statistics and plotted in polar coordinates to show where the most co-activation is concentrated within the walk and run cycles, and, in this study, also to show the instant of co-activation onset on the rostro–caudal maps [% gait cycle] [38,65]:(3)$$\{\begin{array}{c} A=\overset{201}{\underset{i=1}{\sum }}\left(\cos\theta_{t}\times EMG_{i}\right)\\ B=\overset{201}{\underset{i=1}{\sum }}\left(\sin\theta_{t}\times EMG_{i}\right)\\ CoA=\tan^{-1}\left(\frac{B}{A}\right) \end{array}$$
where $\theta_{t}$ is the angle in polar coordinates that varies from 0 to 360°.(v)is the coefficient of multiple correlation (CMC), which measures the overall waveform similarity of a group of curves (the closer to 1 the CMC is, the more similar the curves are) [66,67,68]. The CMC was calculated according to the following formula:(4)$$CMC=\sqrt{1-\frac{(1/(T\left(N-1\right)))\sum_{1}^{N}\sum_{1}^{T}{\left(y_{it}-{\bar{y}}_{t}\right)}^{2}}{(1/\left(\mathrm{TN}-1\right))\sum_{1}^{N}\sum_{1}^{T}{\left(y_{it}-\bar{y}\right)}^{2}}}$$
where T=200 (number of points within the curve), N is the number of curves, $y_{it}$ is the value at the t-th point in the i-th curve, ${\bar{y}}_{t}$ is the average of the two curves at point t, and ${\bar{y}}_{t}$ is the grand mean of all $y_{it}$.

Data over individual strides was calculated for each runner and speed and then averaged across cycles. Within (CMC_Wt*_glob_*, CMC_Wt*_ext_*, CMC_Wt*_flex_*, CMC_Wt*_L_*_3_, CMC_Wt*_L_*_4_, CMC_Wt*_L_*_5_, CMC_Wt*_S_*_1_, CMC_Wt*_S_*_2_) and between (CMC_Bt*_glob_*, CMC_Bt*_ext_*, CMC_Bt*_flex_*, CMC_Bt*_L_*_3_, CMC_Bt*_L_*_4_, CMC_Bt*_L_*_5_, CMC_Bt*_S_*_1_, CMC_Bt*_S_*_2_) runners, the CMC was calculated.

#### 2.4.4. Cross-Correlation

The shape similarity overall in the co-activation maps was evaluated using bidimensional normalized cross-correlation [33,38,69]. In detail, the central value of cross-correlation function R(u,v), with an amplitude ranging from 0 to 1, was used to measure shape similarity between the global co-activation map and the extensor (R*_G-E_*) and flexor (R*_G-F_*) co-activation maps, as well as the shape similarity between the global co-activation map and the rostro–caudal co-activation maps (R*_G-L_*_3_, R*_G-L_*_4_, R*_G-L_*_5_, R*_G-S_*_1_, R*_G-S_*_2_).

#### 2.4.5. Center of Mass Displacement and Spatiotemporal Parameters

The whole-body CoM for each speed and runner was calculated using the “reconstructed pelvis method” [70,71,72], i.e., the geometric center of the triangle formed by the markers over the two anterior superior iliac spines and the sacrum, which is the pelvic center. The displacement in vertical (CoM*_y_*) and mediolateral (CoM*_z_*) directions was then calculated from the COM as the difference between the maximum and minimum values in their respective directions.

The following spatiotemporal parameters were determined for each speed and runner: (i) Toe-off event (TO*_e_*) [% gait cycle]; (ii) stride length [cm]; (iii) stride frequency [Hz]; (iv) foot lift [cm] [33,38,71,73,74]; with the latter calculated as the maximum elevation along the vertical direction (y coordinate) of the centroid formed by the heel, metatarsal head, and lateral malleoli.

#### 2.4.6. Statistical Analysis

A repeated measures ANOVA was used to determine the significance of differences in co-activation parameters computed from global, flexor, extensor, and rostro–caudal maps, as well as the CoM and spatiotemporal parameters, with gait speed as a within-subjects factor (18 levels: from 0.8 to 9.3 km/h), followed by a Bonferroni’s pot-hoc analysis to determine whether there were significant differences between speeds.

An unpaired two-sample *t*-test was used to compare the shape similarity of the global vs extensor co-activation maps and the global vs flexor co-activation maps. A univariate ANOVA was performed with co-activation map shape similarity as a between-group factor (5 levels: global vs each rostro–caudal co-activation map), and a Bonferroni’s post-hoc analysis was performed to assess the significant differences between each shape similarity.

The Watson–Williams test for circular data was applied to CoAs calculated from global, flexor, and extensor co-activation maps, with gait speed as the between-group factor, followed by a Bonferroni’s pot-hoc analysis to see if there were any significant differences between the speeds [33,59,65,75]. Meanwhile, the Harrison–Kanji test for circular data was applied to CoAs calculated from rostro–caudal maps, with gait speed and spinal level as between and within-group factors. A Bonferroni’s post-hoc analysis was used to determine whether there were significant differences between spinal levels and speeds [76].

Gender differences were analyzed using a *t*-test or Mann–Whitney test after verifying the normality of the distributions through a Shapiro–Wilk test.

A partial correlation analysis was performed, excluding the effects of gait speed, between each calculated global, flexor-extensor, and rostro–caudal co-activation map and each CoM and spatiotemporal parameter. Strong correlations were defined as significant correlation coefficients > 0.69 [77].

The significance level was set to 0.05, and all analyses were carried out in MATLAB (8.3.0.532, MathWorks, Natick, MA, USA).

## 3. Results

For each investigated parameter, no significant differences were found based on gender (*p* > 0.05).

### 3.1. Global, Flexor, Extensor, and Rostro–Caudal Co-Activation Maps and Parameters

Figure 1 shows the three-dimensional global map of lower limb muscle co-activation from slow walking to running.

Figure 2 shows the average and standard deviation of 19 runners’ co-activation parameters (CI*_glob_*, Max*_glob_*, FWHM*_glob_*, CMC_Wt*_glob_*_,_ and CMC_Bt*_glob_*) from slow walking to running. A significant main effect of gait speed was found on CI*_glob_*, Max*_glob_*, FWHM*_glob_*, and CMC_Wt*_glob_* (see Table 2). The post-hoc analysis revealed that at the transition between walking (6.8 km/h) and running (7.3 km/h), the values of Ciglob, Max*_glob_*, CMC_Wt*_glob_*, and CMC_Bt*_glob_* were significantly higher, whereas the value of FWHM*_glob_* was significantly lower (see Figure 2 and Table 2).

Figure 3 shows the average and the standard deviation of 19 runners’ CoA*_glob_* from slow walking to running. A significant effect of gait speed was found on CoA*_glob_* (See Table 2). The post-hoc analysis revealed that at the transition between walking (6.8 km/h) and running (7.3 km/h), the values of CoA*_glob_* were significantly lower (see Figure 2 and Table 2).

Figure 4 shows the extensor (left side) and flexor (right side) maps of lower limb muscle co-activation from slow walking to running, with the averages of the TMCf*_ext_* and TMCf*_flex_* curves (black lines), TO*_e_* (black dots), and FWHM*_ext_* and FWHM*_flex_* (the area underlying the TMCf*_ext_* and TMCf*_flex_* red and grey for run and walk, respectively) for all 19 runners. A significant effect of speed was found on extensor and flexor parameters: CI*_ext_* and CI*_flex_*, on the Max*_ext_* and Max*_flex_*, FWHM*_ext_* and FWHM*_flex_*, CMC_Wt*_ext_*, and CMC_Wt*_flex_* (see Table 2). The post-hoc analysis revealed that at the transition between walking (6.8 km/h) and running (7.3 km/h), the values of the CI*_ext_*, Max*_ext_*, and CMC_Bt*_ext_* were significantly higher. In contrast, the value of CMC_Bt*_flex_* was significantly lower (see Table 2).

A significant effect of gait speed was found on CoA*_ext_* and CoA*_flex_* (see Table 2). The post-hoc analysis revealed that at the transition between walking (6.8 km/h) and running (7.3 km/h), the values of CoA*_ext_* were significantly higher. In contrast, the value of CoA*_flex_* was significantly lower (see Table 2).

Figure 5 shows the rostro–caudal (from L3 to S2 spinal level) maps of lower limb muscle co-activation from slow walking to running with the averages of the TMCf*_L_*_3_, TMCf*_L_*_4_, TMCf*_L_*_5_, TMCf*_S_*_1_, TMCf*_S_*_2_ (black lines), TO*_e_* (black dots), and FWHM*_L_*_3_, FWHM*_L_*_4_, FWHM*_L_*_5_, FWHM*_S_*_1_, FWHM*_S_*_2_ (the area underlying the curves, red and grey for run and walk respectively) for all 19 runners.

A significant effect of gait speed was found on CI*_L_*_3_, CI*_L_*_4_, CI*_L_*_5_, CI*_S_*_1_, and CI*_S_*_2_, on the Max*_L_*_3_, Max*_L_*_4_, Max*_L_*_5_, Max*_S_*_1_, and Max*_S_*_2_, on the FWHM*_L_*_3_, FWHM*_L_*_4_, FWHM*_L_*_5_, FWHM*_S_*_1_, FWHM*_S_*_2_, and on the CMC_Wt*_L_*_3_, CMC_Wt*_L_*_4_, CMC_Wt*_L_*_5_, CMC_Wt*_S_*_1_, CMC_Wt*_S_*_2_ (see Table 2). The post-hoc analysis revealed that at the transition between walking (6.8 km/h) and running (7.3 km/h), the values of the CI*_L_*_4_, CI*_L_*_5_, CI*_S_*_1_, CI*_S_*_2_, Max*_S_*_1_, Max*_S_*_2_, CMC_Wt*_S_*_1_, CMC_Bt*_L_*_3_, CMC_Bt*_L_*_4_, CMC_Bt*_S_*_1_, and CMC_Bt*_S_*_2_ were significantly higher. In contrast, the value of the CMC_Bt*_L_*_5_ was significantly lower (see Table 2).

A significant effect of gait speed and level was found on CoA*_L_*_3_, CoA*_L_*_4_, CoA*_L_*_5_, CoA*_S_*_1_, and CoA*_S_*_2_ (see Table 2 and Table 3). The post-hoc analysis on speed revealed that at the transition between walking (6.8 km/h) and running (7.3 km/h), the values of the CoA*_L_*_3_, CoA*_L_*_4_, and CoA*_L_*_5_ were significantly higher. In contrast, the values of the CoA*_S_*_1_ and CoA*_S_*_2_ were significantly lower (see Table 2). The post-hoc analysis on the spine levels revealed that at the transition between walking (6.8 km/h) and running (7.3 km/h), the values of the CoA*_S_*_1_ were significantly higher than the CoA*_L_*_3_, CoA*_L_*_4_, and CoA*_L_*_5_. The values of the CoA*_S_*_2_ were significantly higher than the CoA*_L_*_3_, CoA*_L_*_4_, and CoA*_L_*_5_ (see Table 3).

### 3.2. Cross-Correlation

A significantly higher shape similarity between the global and the extensor co-activation map compared to the global and flexor co-activation map was found (see Table 4).

A significant effect of shape similarity was found between the global and rostro–caudal co-activation maps (see Table 4). The post-hoc analysis revealed that the value of the shape similarity between the global and S1 co-activation maps was significantly higher than the global and L3, L5, and S2 co-activation maps. The shape similarity between the global and L4 co-activation maps was significantly higher than the global, L3, and S2 co-activation maps (see Table 4).

### 3.3. Center of Mass Displacement and Spatiotemporal Parameters

Figure 6 shows three-dimensional CoM maps in the vertical (CoM*_y_*) and mediolateral (CoM*_z_*) directions. The maps were created using the CoM*_y_* and CoM*_z_* displacement average curves of 19 runners calculated on the gait cycle, from 0% to 100% (x-axis), for each speed of walking and running performed, from 0.8 to 9.3 km/h (y-axis), and with a variable amplitude (z-axis). A significant effect of the speed was found on the CoM*_y_* and the CoM*_z_*. The post-hoc analysis revealed that at the transition between walking (6.8 km/h) and running (7.3 km/h), the values of the CoM*_y_* displacement were significantly higher. In contrast, the values of the CoM*_z_* displacement were significantly lower (see Figure 6 and Table 2).

A significant effect of the speed was found on the TO*_e_*, stride length, stride frequency, and foot lift. The post-hoc analysis revealed that the values of the stride length, foot lift, and stride frequency were significantly higher. In contrast, the values of the TO*_e_* were significantly lower at the transition between walking (6.8 km/h) and running (7.3 km/h) (see Figure 7 and Table 2).

### 3.4. Correlations

Regardless of the gait speed, the CoM_y_ was positively correlated with a higher CI*_glob_* (*r* = 0.88, *p* < 0.001) and Max*_glob_* (*r* = 0.89, *p* < 0.001) and negatively correlated with the FWHM*_glob_* (*r* = −0.83, *p* < 0.001) values; whereas, the CoM_z_ was positively correlated with the CoA*_glob_* values (*r* = 0.97, *p* < 0.001). The stride length values were negatively correlated with the CI*_glob_* (*r* = −0.98, *p* < 0.001) and Max*_lob_* (*r* = −0.99, *p* < 0.001) values and positively correlated with the FWHM*_glob_* (*r* = 0.80, *p* < 0.001). The foot lift values were positively correlated with the Max*_glob_* values (*r* = 0.72, *p* = 0.001), and the cadence values were negatively correlated with the CoA*_glob_* values (*r* = −0.72, *p* < 0.001). In contrast, TOe values were positively correlated with CoA*_glob_* values (*r* = 0.79, *p* < 0.001).

## 4. Discussion

The objective of this study was to explore the behavior of the lower limb muscles’ co-activation across several speeds, ranging from walking to running, as an expression of the CNS’s global strategy for controlling the simultaneous activation of several lower limb muscles during locomotion.

We found that at higher speeds, the global co-activation index rises with increasing co-activation function values and that co-activation occurs earlier and for a shorter period than at slower speeds. This is especially evident while transitioning from walking to running, as indicated by higher CI*_glob_* and Max*_glob_* values when walking and lower CoA*_glob_* and FWHM*_glob_* values while running (see Figure 2 and Figure 3). We also found that running resulted in increased vertical displacement, foot lift, stride frequency, and decreased lateral displacement and stride length, compared to walking (see Figure 6 and Figure 7), which is consistent with earlier research on the biomechanics of running [19,20,21,22]. Interestingly, regardless of gait speed, global co-activation occurring earlier and in a confined portion of the gait cycle was positively correlated with higher vertical displacement, stride frequency, and foot lift and negatively correlated with lower lateral displacement and stride length.

From slow walking to running, we found a different set of curve shapes in lower limb muscle co-activation (see Figure 1). The time-varying multi-muscle co-activation function revealed a one-hump configuration during slow walking, a four-hump configuration during fast walking, and a three-hump configuration during running. The lonely hump we found during slow walking, as well as the first hump we discovered during running, paralleled the entire duration of the stance contact phase. In contrast, the first two humps of fast walking distinctly corresponded to the loading (in the range of 10–20% of the gait cycle) and push-off (in the range of 40–60% of the gait cycle) subphases, respectively.

Overall, these humps reflected an increase in whole-limb stiffness in response to ground impact for weight acceptance and pushing on the ground for propulsive purposes, which was consistent with previous findings that the spatiotemporal profile of global co-activation matches that of ground reaction force (GRF) [33,38]. Both gait variability and CoM oscillation are known to increase during slow walking, resulting in less stable walking compared to faster gait [9,78,79]. The longer duration, lower magnitude (see Figure 1), and higher within and between subject variability of lower limb global co-activation observed at slow walking, as expressed by CMC values (See Figure 2), suggest that the increase in whole-limb stiffness is exerted within a long-lasting and variable “safety strategy,” primarily designed to maintain dynamic balance during body progression [78,79].

Conversely, the shorter duration, higher magnitude (see Figure 1), and lower within and between-subject variability of lower limb global co-activation (see Figure 2) observed during running suggest that whole-limb stiffness is exerted by more synchronized and stable muscle activation. It also implies that when transitioning from walking to running, the increased whole-limb muscle co-activation during the foot contact phase is intended to absorb ground impact and generate propulsive forces in a single motor act, most likely within a unique motor strategy. As a result, the CI could be a useful index for reflecting and assessing the efficiency of the “leg-spring” stiffness mechanism, which is correlated with more economical running [80,81].

The findings of the partial correlation analysis support the observed mechanisms of global co-activation. Regardless of gait speed, earlier and shorter values of global co-activation during the gait cycle reduce the stride length while increasing vertical and decreasing lateral CoM displacements. As a result, rather than being the result of changes in gait speed, it is possible to hypothesize that the observed differences in spatiotemporal and kinematic variables between walking and running could be the result of global co-activation, which is a sensory-control integration process used by the CNS to deal with a more demanding and potentially unstable task like running [82,83,84]. This explains why similar results in terms of co-activation indexes and kinematic characteristics were found in subjects with abnormal gait stability control, such as cerebellar ataxia, who attempt to compensate for gait instability by using global co-activation [33,71,73].

When we analyzed the co-activation of either flexor or extensor muscles separately, we found that the function curve of the extensor muscles matched that of the whole limb, whereas that of the flexor muscles clearly differed from it (see Figure 1 and Figure 4). These findings suggest that the main contribution to the whole limb stiffening during walking and running is mainly given by the co-activation of the extensor muscles according to their role in weight acceptance and propulsive function [85]. Such extensor activation clearly corresponds to the first hump of the global co-activation of both running and walking (both slow and fast) curves, as well as to the second hump of fast walking. Conversely, the co-activation curve of the flexor muscles increased during the early step air phase in line with the role of flexor muscles in limb lifting [85]. The co-activation of the flexor muscles, although small, was clearly present during running as the second hump of the global co-activation curve and during fast walking as the third hump (see Figure 4). This last result is of particular interest because it suggests that the flexor muscles are a key factor in transitioning from fast walking to running. It also fits well with the observation of a propagation delay in muscle co-activation from L3 to S2 (see Figure 5). These findings are in line with previous findings on walking [41,86,87] and reinforce the hypothesis that the co-activation of the flexor muscles (e.g., rectus femoris and iliopsoas), whose motoneurons are located more rostrally within the spinal cord, may play a crucial role in the switch from walking to running. Interestingly, although there was a clear progressive increase in the amount of co-activation of either global extensors or flexors from slow to fast walking, we found an increase in flexor co-activation just before the running speed threshold in the late fast walking trials, followed by a mild reduction in co-activation in the early running trials (see Figure 4 and Table 2). One of the key features in the transition from fast walking to running is the optimal exploitation of the mass-spring mechanism to generate kinetic energy during running [15,23]. We might infer that such an advantage, which occurs during ground contact, is related to global co-activation and reduces the need to coactivate the flexor muscles during the step air phase in early running compared to late fast walking. It is important to note that the co-activation of the flexor muscles corresponded to a small peak (second hump) in the “hollow” of the global co-activation curve, possibly suggesting that the flexor muscles need to be coactivated when the limb stiffness is reduced to facilitate the forward progression.

This study has several limitations. First, due to the experimental setup, anteroposterior CoM displacements could not be assessed, and thus, the kinetic energy parameters could not be calculated. We also did not account for oxygenation parameters, so our data cannot provide a complete characterization of global co-activation on running economy. However, because greater neuromuscular activation, vertical stiffness, and the ability to rapidly produce force throughout the lower limb during ground contact have been shown to correlate with more economical running [81,88,89], it is expected that CI values may correlate with the energetic running profile in future studies evaluating running performances. Other limitations are related to the superimposition of gait speeds through the use of a treadmill, which might have affected some gait parameters [38,74], and the lack of re-recording of the participants’ training programs. Therefore, further studies examining global co-activation indices during overground running are needed.

## 5. Conclusions

The findings of this study revealed a distinct pattern in global lower limb muscle co-activation across gait speeds and between walking and running tasks, with higher speeds resulting in earlier and shorter durations. This mechanism implies that when running, subjects employ a single motor strategy to absorb ground impact and generate propulsive forces, making the CI a potentially useful index to reflect the efficiency of the “leg-spring” stiffness mechanism and, as a result, to assess running efficiency. Focusing on muscle co-activation efficiency may assist sports professionals in tailoring training programs to improve running efficiency, as well as monitoring recovery after injury.

## Figures and Tables

**Figure 1 bioengineering-11-00288-f001:**
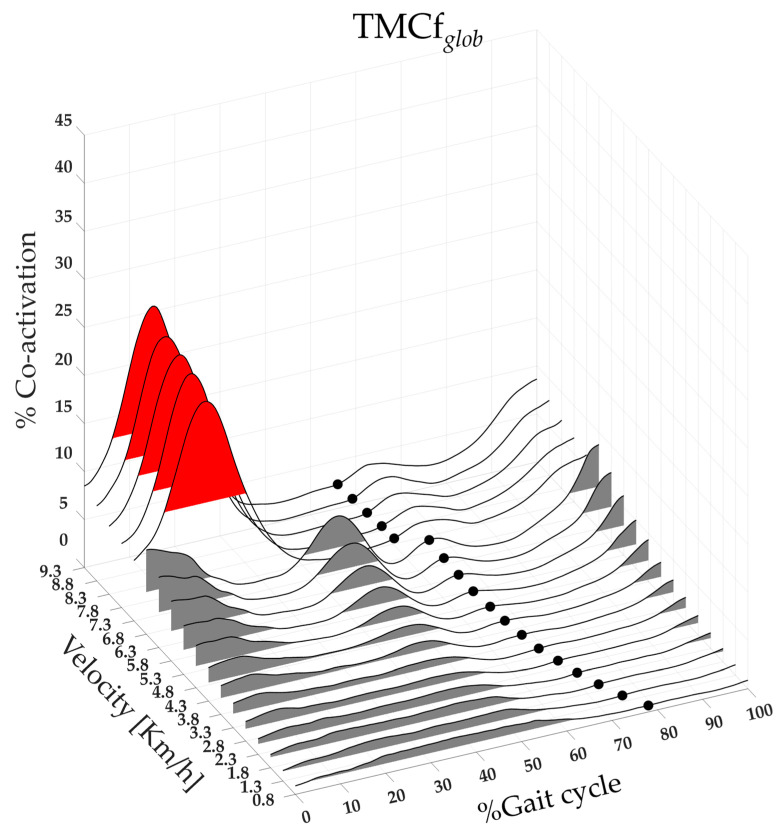
Three-dimensional global map of lower limb muscle co-activation from slow walking to running. Each curve represented the average TMCf*_glob_* curves (black line) of 19 runners calculated on the whole gait cycle, from 0% to 100% (x-axis), for each speed of walking and running performed, from 0.8 to 9.3 km/h (y-axis), and with amplitudes ranging from 0% to 100% of co-activation (z-axis). The TO*_e_* (black dots) and FWHM*_glob_* (the area underlying the TMCf*_glob_*, grey and red area for walk and run velocity, respectively) for each speed as a mean between all runners are also shown on the map. TMCf*_glob_:* global time-varying multi-muscle co-activation function; TO*_e_:* Toe-off event; FWHM*_glob_:* full width at half maximum of the global co-activation.

**Figure 2 bioengineering-11-00288-f002:**
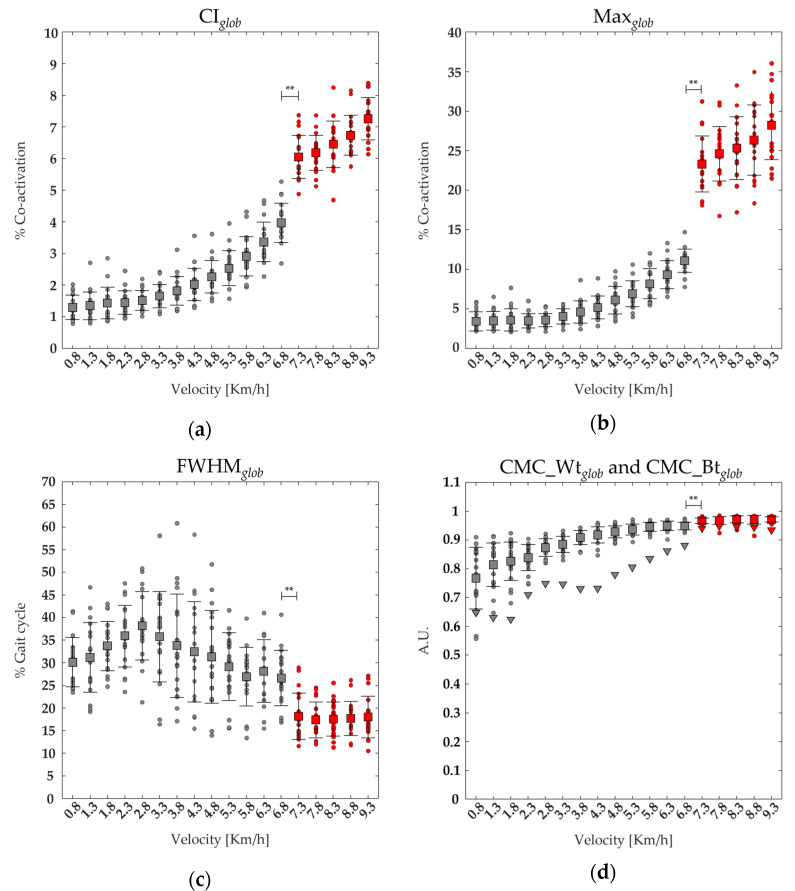
The average (squares) and standard deviation (black bars) of 19 runners’ (dot) global co-activation parameters ranging from slow walking (gray) to running (red) are reported: (**a**) average of the global co-activation level [% co-activation] (CI*_glob_*,), (**b**) maximum value of the global co-activation (Max*_glob_*), (**c**) the full width at half maximum of global co-activation (FWHM*_glob_*), and (**d**) coefficient of multiple correlation within runners of global co-activation (CMC Wt*_glob_*). Triangles turned (**d**) represent the coefficient of multiple correlation of global co-activation between runners (CMC Bt*_glob_*). ** statistical significance (*p* < 0.001).

**Figure 3 bioengineering-11-00288-f003:**
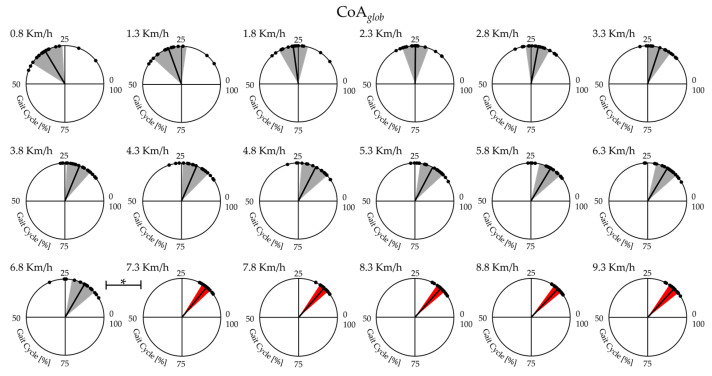
The center of activity (CoA*_glob_*) of the lower limb muscle co-activation curves from slow walking (in gray color area from 0.8 to 6.8 km/h) to running (red color area, from7.3 to 9.3 km/h)with respective velocities in the top left. Each dot in the circumference represents a single subject’s mean CoA value. In contrast, the mean value and SD of the CoA of all subjects are represented by the solid line and the width of the circular sector, respectively. * statistical significance (*p* < 0.05).

**Figure 4 bioengineering-11-00288-f004:**
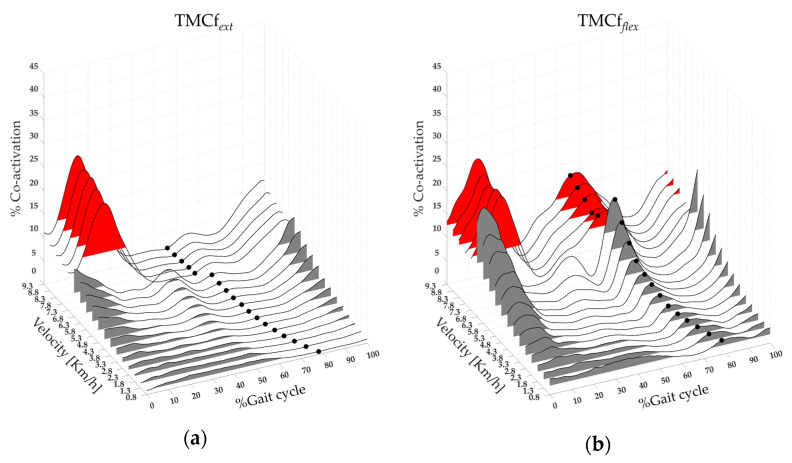
Three-dimensional extensor (**a**) and flexor (**b**) maps of the lower limb muscle co-activation from slow walking to running (from 0.8 to 9.3 km/h). Each curve (black line) represents the average co-activation of extensor (TMCf*_ext_* (**a**)) and flexor (TMCf*_flex_* (**b**)) muscles of 19 runners. Black dots (**a**,**b**) represent the Toe-off event, and the grey and red areas represent the full width at half maximum of extensor (**a**) and flexor (**b**) muscle co-activation for walk and run velocity, respectively.

**Figure 5 bioengineering-11-00288-f005:**
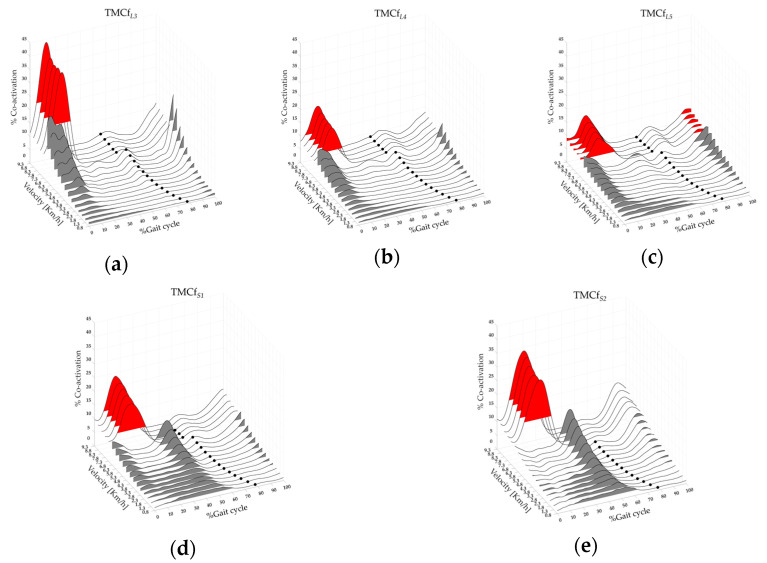
Three-dimensional rostro–caudal (from L3 to S2 spinal level) maps of the lower limb muscle co-activation from slow walking to running (from 0.8 to 9.3 km/h): (**a**) L3 co-activation (TMCf*_L_*_3_), (**b**) L4 co-activation (TMCf*_L_*_4_), (**c**) L5 co-activation (TMCf*_L_*_5_), (**d**) S1 co-activation (TMCf*_S_*_1_), (**e**) S2 co-activation (TMCf*_S_*_2_). Each curve represented the average TMCf*_L_*_3_ (**a**), TMCf*_L_*_4_ (**b**), TMCf*_L_*_5_ (**c**), TMCf*_S_*_1_ (**d**), and TMCf*_S_*_2_ (**e**) curves (black line) of 19 runners. The black dots represent the Toe-off event, and the grey and red areas represent the full width at half maximum of L3 (**a**), L4 (**b**), L5 (**c**), S1 (**d**), and S2 (**e**) co-activations for walk and run velocity respectively.

**Figure 6 bioengineering-11-00288-f006:**
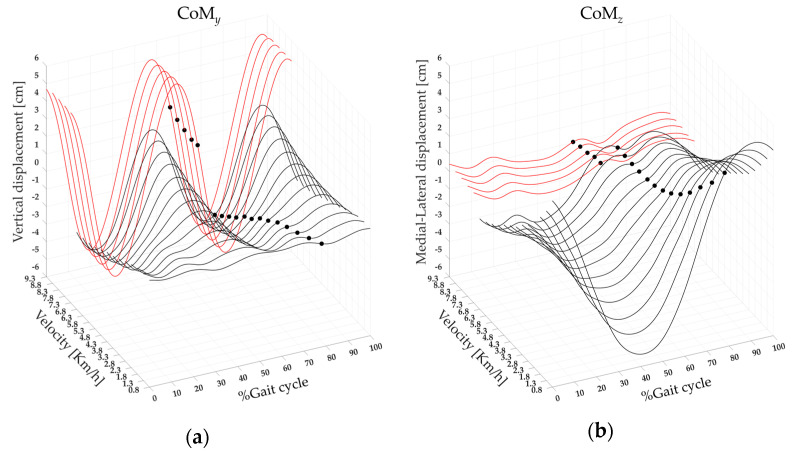
The three-dimensional CoM maps from slow walking to running in the vertical (CoM*_y_*, (**a**)) and mediolateral (CoM*_z_*, (**b**)) directions. Each curve represented the average CoM*_y_* and CoM*_z_* (black line) curves of 19 runners, as well as the Toe−off event (black dots). The average (squares) and standard deviation (black bars) of 19 runners’ (dot) CoM*_y_* (**c**) and CoM*_z_* (**d**) displacements, ranging from slow walking (gray) to running (red). ** statistical significance (*p* < 0.001).

**Figure 7 bioengineering-11-00288-f007:**
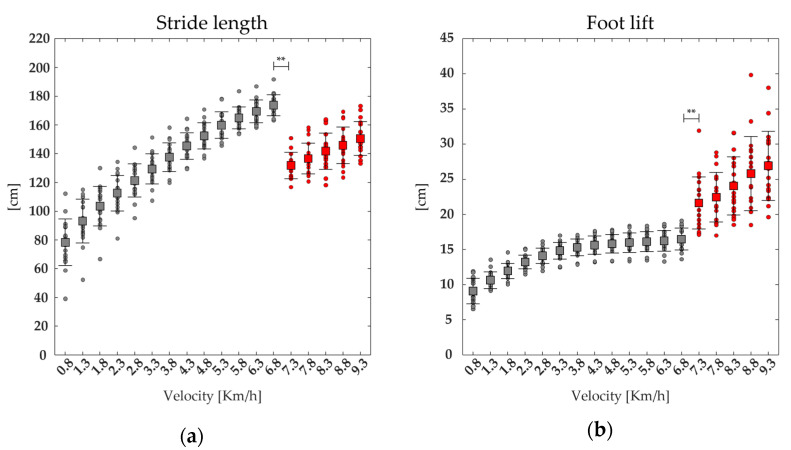
The average (squares) and standard deviation (black bars) of 19 runners’ (dot) spatiotemporal parameters: stride length (**a**), foot lift (**b**), stride frequency (**c**), and Toe−off event (TO*_e_*, (**d**)), ranging from slow walking (gray) to running (red). ** statistical significance (*p* < 0.001).

**Table 1 bioengineering-11-00288-t001:** Each dot in the table indicates muscles included in the time-varying co-activation (TMCf) function for each muscle co-activation investigated: global (all the monitored lower limb muscles), extensor (extensor muscle subgroup), flexor (flexor muscle subgroup), and rostro–caudal organization (muscles on the basis of their spinal segment of innervation, see text for further details). Smallest dots indicate a halved weight (amplitude of muscle activity multiplied by 0.5) for that specific muscle in the TMCf function.

		Maps							
		Global	Extensor	Flexor	L3	L4	L5	S1	S2
Muscles	GM	⬤	⬤			⬤	⬤	⬤	
	RF	⬤		⬤	⬤	⬤			
	VL	⬤	⬤			⬤			
	VM	⬤	⬤		⬤	⬤			
	TFL	⬤		⬤	⬤	⬤	⬤	⬤	
	ST	⬤	⬤			•	⬤	⬤	⬤
	BF	⬤	⬤				●	●	⬤
	TA	⬤		⬤		⬤	⬤	⬤	
	GasM	⬤	⬤					⬤	⬤
	GasL	⬤	⬤					⬤	⬤
	SOL	⬤					●	⬤	⬤
	P	⬤				●	⬤	⬤	

GM: Gluteus medius, RF: rectus femoris, VL: vastus lateralis, VM: vastus medialis, TFL: tensor fascia latae, ST: semitendinosus, BF: biceps femoris, TA: tibialis anterior, GasM: gastrocnemius medialis, GasL: gastrocnemius lateralis, SOL: soleus, P: peroneus longus.

**Table 2 bioengineering-11-00288-t002:** Main effect and post-hoc comparisons at the transition between walking (6.8 km/h) and running (7.3 km/h), with corresponding values, for each co-activation parameters of each for each muscle co-activation studied: global (glob), extensor (ext), flexor (flex), and rostro–caudal organization (L3, L4, L5, S1, S2).

Parameters	Main Effect Velocity		Post-Hoc Velocity Transition		
	*F_(df)_*	*p*	Value at 6.8 km/h (Mean ± Std)	Value at 7.3 km/h (Mean ± Std)	*p* Value
CI*_glob_*	F_(1,17)_ = 641.04	<0.001	3.97 ± 0.62	6.05 ± 0.69	<0.001
CI*_ext_*	F_(1,17)_ = 388.04	<0.001	3.82 ± 0.83	6.53 ± 0.99	<0.001
CI*_flex_*	F_(1,17)_ = 240.06	<0.001	9.03 ± 1.72	10.3 ± 2.4	/
CI*_L_*_3_	F_(1,17)_ = 137.28	<0.001	6.21 ± 2.06	8.09 ± 2.28	/
CI*_L_*_4_	F_(1,17)_ = 409.77	<0.001	3.85 ± 0.71	5.22 ± 0.72	<0.001
CI*_L_*_5_	F_(1,17)_ = 351.04	<0.001	4.22 ± 0.72	5.87 ± 0.78	<0.001
CI*_S_*_1_	F_(1,17)_ = 461.13	<0.001	5.14 ± 0.70	6.88 ± 0.91	<0.001
CI*_S_*_2_	F_(1,17)_ = 464.98	<0.001	5.83 ± 1.26	9.24 ± 1.5	<0.001
Max*_glob_*	F_(1,17)_ = 321.71	<0.001	11.08 ± 1.48	23.30 ± 3.54	<0.001
Max*_ext_*	F_(1,17)_ = 152.36	<0.001	12.32 ± 2.77	24.42 ± 5.48	<0.001
Max*_flex_*	F_(1,17)_ = 104.01	<0.001	30.16 ± 8.91	26.51 ± 7.28	/
Max*_L_*_3_	F_(1,17)_ = 76.83	<0.001	31.5 ± 14.27	40.2 ± 14.41	/
Max*_L_*_4_	F_(1,17)_ = 156.75	<0.001	11.93 ± 3.75	14.47 ± 4.13	/
Max*_L_*_5_	F_(1,17)_ = 93.91	<0.001	11.7 ± 2.33	15.18 ± 3.5	/
Max*_S_*_1_	F_(1,17)_ = 192.85	<0.001	16 ± 3.07	24.64 ± 4.39	<0.001
Max*_S_*_2_	F_(1,17)_ = 189.34	<0.001	18.91 ± 4.04	34.19 ± 6.3	<0.001
FWHM*_glob_*	F_(1,17)_ = 29.31	<0.001	26.62 ± 6.08	18.17 ± 5.14	0.01
FWHM*_ext_*	F_(1,17)_ = 9.31	<0.001	21.15 ± 6.3	20.07 ± 6.7	/
FWHM*_flex_*	F_(1,17)_ = 11.13	<0.001	20.74 ± 6.66	31.14 ± 8.65	/
FWHM*_L_*_3_	F_(1,17)_ = 11.37	<0.001	14.45 ± 3.81	15.69 ± 3.39	/
FWHM*_L_*_4_	F_(1,17)_ = 20.28	<0.001	16.86 ± 4.58	18.32 ± 4.6	/
FWHM*_L_*_5_	F_(1,17)_ = 8.22	<0.001	25.45 ± 7	29.33 ± 7.67	/
FWHM*_S_*_1_	F_(1,17)_ = 13.48	<0.001	20.68 ± 6.17	20.53 ± 5.81	/
FWHM*_S_*_2_	F_(1,17)_ = 4.06	<0.001	20.34 ± 7.9	20.08 ± 5.65	/
CoA*_glob_*	F_(1,17)_ = 24.96	<0.001	16.74 ± 5.38	14.05 ± 2.12	<0.01
CoA*_ext_*	F_(1,17)_ = 22.95	<0.001	9.1 ± 4.09	12.76 ± 2.88	<0.001
CoA*_flex_*	F_(1,17)_ = 8.74	<0.001	97.18 ± 11.8	15.63 ± 13.91	<0.001
CoA*_L_*_3_	F_(4,17)_ = 58.01	<0.001	7.31 ± 2.58	13.2 ± 2.28	<0.001
CoA*_L_*_4_			5.96 ± 2.93	11.63 ± 2.31	<0.001
CoA*_L_*_5_			5.72 ± 5.31	8.89 ± 3.82	<0.01
CoA*_S_*_1_			30.35 ± 4.3	15.2 ± 2.17	<0.001
CoA*_S_*_2_			28.01 ± 5.1	15.88 ± 2.98	<0.001
CMC_Wt*_glob_* (CMC_Bt*_glob_*)	F_(1,17)_ = 54.38	<0.001	0.95 ± 0.01 (0.86)	0.97 ± 0.01 (0.88)	0.02
CMC_Wt*_ext_* (CMC_Bt*_ext_*)	F_(1,17)_ = 44.98	<0.001	0.95 ± 0.02 (0.86)	0.96 ± 0.01 (0.89)	/
CMC_Wt*_flex_* (CMC_Bt*_flex_*)	F_(1,17)_ = 55.77	<0.001	0.93 ± 0.04 (0.79)	0.92 ± 0.03 (0.75)	/
CMC_Wt*_L_*_3_ (CMC_Bt*_L_*_3_)	F_(1,17)_ = 24.31	<0.001	0.97 ± 0.02 (0.81)	0.96 ± 0.02 (0.90)	/
CMC_Wt*_L_*_4_ (CMC_Bt*_L_*_4_)	F_(1,17)_ = 42.25	<0.001	0.96 ± 0.01 (0.87)	0.96 ± 0.01 (0.92)	/
CMC_WtL5 (CMC_Btglob)	F_(1,17)_ = 38.82	<0.001	0.93 ± 0.02 (0.84)	0.92 ± 0.03 (0.83)	/
CMC_WtS1 (CMC_BtS1)	F_(1,17)_ = 49.56	<0.001	0.94 ± 0.01 (0.86)	0.96 ± 0.01 (0.91)	0.04
CMC_WtS2 (CMC_BtS2)	F_(1,17)_ = 48.76	<0.001	0.94 ± 0.02 (0.85)	0.95 ± 0.02 (0.89)	0.02
CoM*y*	F_(1,17)_ = 426.2	<0.001	6.60 ± 0.83	10.99 ± 1.92	<0.001
CoM*z*	F_(1,17)_ = 120.29	<0.001	4.57 ± 1.23	2.78 ± 0.88	<0.001
TO*e*	F_(1,17)_ = 940.64	<0.001	62.57 ± 0.70	57.91 ± 1.06	<0.001
stride length	F_(1,17)_ = 253.03	<0.001	173.78 ± 7.25	131 ± 9.24	<0.001
stride frequency	F_(1,17)_ = 714.22	<0.001	1.13 ± 0.05	1.31 ± 0.07	<0.001
foot lift	F_(1,17)_ = 108.03	<0.001	16.47 ± 1.57	21.63 ± 3.69	<0.01

CI: synthetic co-activation index; Max: maximum value of the co-activation; FWHM: full width at half maximum of the co-activation; CoA: center of activity of the co-activation; CMC_W and CMC_B: the coefficient of multiple correlations between and within runners; CoM*y* and CoM*z*: center of mass displacement in vertical and mediolateral direction; TO*e*: Toe-off event.

**Table 3 bioengineering-11-00288-t003:** Main effect and post-hoc comparisons of each spinal level (from L3 to S2) at walking (6.8 km/h) and running (7.3 km/h), with corresponding values for the lower limb muscle co-activation maps.

Main Effect Level		Velocity	Post-Hoc Level					
*F_(df)_*	*p*			CoA*_L_*_3_	CoA*_L_*_4_	CoA*_L_*_5_	CoA*_S_*_1_	CoA*_S_*_2_
F_(4,17)_ = 511.50	<0.001	6.8 km/h	CoA*_L_*_3_	/				
			CoA*_L_*_4_	<0.01	/			
			CoA*_L_*_5_	0.01	0.04	/		
			CoA*_S_*_1_	<0.001	<0.001	<0.001	/	
			CoA*_S_*_2_	<0.001	<0.001	<0.001	<0.01	/
		7.3 km/h	CoA*_L_*_3_	/				
			CoA*_L_*_4_	<0.01	/			
			CoA*_L_*_5_	<0.001	<0.001	/		
			CoA*_S_*_1_	<0.01	<0.001	<0.001	/	
			CoA*_S_*_2_	<0.001	<0.001	<0.001	<0.001	/

CoA: center of activity of the co-activation.

**Table 4 bioengineering-11-00288-t004:** Statistical significance (and corresponding mean and standard deviation values) of the shape similarity between the global and the extensor co-activation map (*R_G-E_*) as well as the global and flexor (*R_G-F_*) co-activation map. Main effect and post-hoc comparisons (and corresponding mean and standard deviation values) of shape similarity between the global and rostro–caudal (from L3 to S2 spinal level: *R_G-L_*_3_, *R_G-L_*_4_, *R_G-L_*_5_, *R_G-S_*_1_, *R_G-S_*_2_) co-activation maps.

	Shape Similarity (Mean ± Std)	*t*-Test *p*							
*R_G-E_*	0.97 ± 0.02	<0.001							
*R_G-F_*	0.70 ± 0.12								
		Main effect shape similarity		Post-hoc shape similarity					
		*F_(df)_*	*p*		*R_G-L_* _3_	*R_G-L_* _4_	*R_G-L_* _5_	*R_G-S_* _1_	*R_G-S_* _2_
*R_G-L_* _3_	0.88 ± 0.08	F_(1,4)_ = 11.23	<0.001	*R_G-L_* _3_	/				
*R_G-L_* _4_	0.93 ± 0.03			*R_G-L_* _4_	0.02	/			
*R_G-L_* _5_	0.89 ± 0.04			*R_G-L_* _5_	/	/	/		
*R_G-S_* _1_	0.96 ± 0.02			*R_G-S_* _1_	<0.001	/	<0.01	/	
*R_G-S_* _2_	0.87 ± 0.06			*R_G-S_* _2_	/	<0.01	/	<0.001	/

## Data Availability

The raw data supporting the conclusions of this article will be made available by the authors on request.

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
