# Peer review of "The Lower Limb Muscle Co-Activation Map during Human Locomotion: From Slow Walking to Running"

_bioengineering, 2024, doi:10.3390/bioengineering11030288_

Round 1

Reviewer 1 Report

Comments and Suggestions for Authors

This article explores the coordinated activation of lower limb muscles at various speeds, ranging from walking to running, as an expression of the central nervous system's global strategy in controlling the simultaneous activation of multiple lower limb muscles during the process of movement. Research in this area may contribute to a better understanding of how the central nervous system regulates and coordinates the activity of multiple lower limb muscles at different movement speeds, adapting to various exercise requirements. However, there are still some points that need correction, primarily in the methodology section. Here below are specific comments:

1. Has the number of subjects been scientifically validated (e.g., through G*power calculations) or referenced from similar studies? If so, please provide the reference(s) after the sentence.

2. Is the formulation of the inclusion criteria for the participants based on relevant literature?

3. In the "Data Acquisition" section, the process of collecting surface electromyography (sEMG) is not detailed enough. Typically, when collecting electromyographic data, the areas to be sampled from the subjects are usually shaved to ensure optimal conductivity and reduce potential interference from the experimental environment.

4. lines 117-118: The reference list appears in two different formats. Is there a lack of uniform formatting?

5. Lines 126-127: Regarding the choice of the camera, I noticed that you also haven't provided relevant explanations or included any references. Therefore, please briefly explain why a video resolution of 640x480 pixels was chosen. This may include considerations for balancing image quality and file size, as well as its relevance to the experimental objectives. Additionally, if available, provide information on the environmental conditions during video recording, such as lighting conditions, shooting angles, etc. This can help readers understand the background conditions of data collection.

6. Line 134: Describe how the data related to heel strike and toe-off events were handled during analysis, as well as the rationale behind the use of the polynomial method for time normalization. This will aid readers in understanding the specific methodologies employed in the study and allow for an assessment of their validity.

7. Line 151: The mention in the text of introducing the processed electromyographic (EMG) signals into a time-dependent function with S-shaped weights raises the question: What is the purpose of choosing this function? What advantages does it have compared to other functions? Could you introduce the specific parameters of this function: sigmoid-weighted time-dependent function?

8. Please provide a detailed description of Table 1, including the determination of the extensor and flexor muscles, and how muscle subgroups are utilized for assessment. Clarifications are needed for terms such as " rostro-caudal organization", " subgroups of muscles ", and " concentric function ".

9. Line 170: This sentence shouldn't have a two-character indentation for the first line, correct?

10. Please verify the formatting of the table to ensure its adherence to the appropriate table format. Additionally, it is advisable to include footnotes in tables as a standard practice.

11. line 535: The reference formatting is not consistent; for instance, the year of publication needs to be bolded.

Comments on the Quality of English Language

no

Author Response

We would like to thank the Reviewer for the time spent on our manuscript and her/his qualified comments and suggestions. In the attached file, our answers (in underlined italic style) and the revised parts of the manuscript (between apices in underlined italic style here and in blue in the text). Furthermore, a minor editing of English language was performed as required.

Reviewer 2 Report

Comments and Suggestions for Authors

Dear authors,

This is great work, with a scientifically sound introduction, a robust methodology and a satisfying discussion. The results are numerous and perhaps you could consider removing Tables 2-3-4 to a supplementary section to make it easier for readers.

Below you can find some minor comments and corrections:

Line 106: better to change "acclimated" to "familiarized".

Lines 107-108: This part is not very clear to me. So, the subject is on the treadmill and starts walking for instance at 1,3km/h for 30 sec, stops and then rests for 1 min, then starts running at 7,3km/h for another 30 sec, rests for 1min etc? How did he rest, was he on the treadmill, standing still or what? Also, at the end of each 30s of trial duration period, whatever treadmill velocity was on dropped to 0? Then, at the next trial, the subject started moving right away at the next velocity? Please clarify. 

Line 116: It is suggested to use 'SOL" for soleus as the acronym typically found in studies.

Lines 146-147: mean of 3 largest peak values regardless of movement speed? On one hand it appears wrong since speed varied significantly during  the walking and the running trials, on the other hand do the authors consider this amplitude normalization as a global one or something similar? Please clarify.

Line 148: Burden 2010 reference is missing.

Lines 238-242: what is the within-subjects factor in this RM Anova model then? I would imagine that the various gait speeds is then within-subjects factor.

Lines 272-277: It is suggested that this information be added to the legend of Figure 1. This is not results, rather a full description of the Figure.

Lines 308-331, Figure 3: It is suggested to add measurement units at each velocity in the Figure, and to clarify in the Legend where the velocity values are shown in the fiure (i.e., top left of each plot).

Line 429: remove the duplicate word "co-activation".

Line 474: The authors suggest to look at Fig.4 to observe the differences in the coactivation pattern between extensor and flexor muscles. Do they mean to visually compare between Fig,4 and Fig.1 which shows the global coactivation pattern? If so, please clarify it in the text.

Line 481: correct to "...in line with the role of the flexor muscles...".

Line 483: correct to "particular".

Line 494: it is suggested to replace "passage" with "transition".

Lines 592-593: The doi is not a match for this paper. Please correct.

Line 694: the reference here is not complete, please correct.

Comments on the Quality of English Language

No comments here. Please see comments above.

Author Response

(The authors gave the same response as above.)

Reviewer 3 Report

Comments and Suggestions for Authors

The lower limb muscle co-activation map during human loco-motion: from slow walking to running.

1. Abstract: Abstract lacks statistical indexes. Authors stated that significant differences were found between parameters, but no p and F values are presented.

2. Introduction:

Authors should provide more details regarding the connection between time-varying multi-muscle coactivation function and the CNS's function.

What is the importance of the study? What will be the possible benefit for coaches and strength and conditioning professionals from the particular study?

3. Methods: what were the instructions regarding training, nutrition and rest that participants had prior to measurement?

 Please, provide a sample power analysis.

 There should be an intra-class correlation coefficient for all measurements.

4. Results: Tables should be self-explanatory. Add the footnotes to tables. Also, see if this as applicable to figures as well.  

5. Discussion: Is there a significant difference between male and female regarding the gait measurement and variables?

Also, how safe are these results especially in a sample that consisted from both male and female participants?

Add a conclusion paragraph with a take home message.

Author Response

(The authors gave the same response as above.)

Round 2

Reviewer 1 Report

Comments and Suggestions for Authors

The reviewer recognized the authors for their efforts in revision. The quality of the article has improved dramatically. The reviewer thinks this article can be accepted after one minor revision.

1. As highlighted in the article, current research is focused on moving from slow walking to running. The reviewers recognize this: the importance of gait in human daily life. It is of great interest to explore lower limb muscle co-activation patterns in response to changes in gait patterns from slow walking to running. This is a significant study. However, the Introduction section is rather lacking in gait, which results in the inability to introduce the topic of gait patterns well. Therefore, the author can add the relevant gait pattern description in the third paragraph (Lines 57-58) to complete the overall logical content. Also, this requires citing the most recent literature on gait pattern research: (1) A new method proposed for realizing human gait pattern recognition: inspirations for the application of sports and clinical gait analysis; (2) Explaining the differences of gait patterns between high and low-mileage runners with machine learning. 

2. Lines 191-192: As mentioned by the authors, for the sigmoid-weighted time-dependent function, is this the muscle activation solution function? I guess I understand what the author is trying to say. In general, the solution process for muscle activation would use a nonlinear model. For example, previous research has used the recursive model (second-order differential equation) to solve the muscle activation by the obtained normalized signal. The authors can refer to the following related studies: https://doi.org/10.1016/j.cmpb.2023.107848; https://doi.org/10.1016/j.cmpb.2023.107761. It includes processing such as filtering EMG signal data, which can better illustrate the feasibility and effectiveness of the method by referring to the related studies.

Comments on the Quality of English Language

no

Author Response

We would like to thank the Reviewer for the time spent on our manuscript and her/his qualified comments and suggestions. In the following, our answers (in underlined italic style) and the revised parts of the manuscript (between apices in underlined italic style here and in blue in the text).

Reviewer 3 Report

Comments and Suggestions for Authors

No comment

Author Response

We would like to thank the Reviewer for the time spent on our manuscript.